# A Simple Contrastive Learning Objective for Alleviating Neural Text Degeneration

## Abstract

The cross-entropy objective has proved to be an all-purpose training objective for autoregressive language models (LMs). However, without considering the penalization of problematic tokens, LMs trained using cross-entropy exhibit text degeneration. To address this, unlikelihood training has been proposed to reduce the probability of unlikely tokens predicted by LMs. But unlikelihood does not consider the relationship between the label tokens and unlikely token candidates, thus showing marginal improvements in degeneration. We propose a new *contrastive token* learning objective that inherits the advantages of cross-entropy and unlikelihood training and avoids their limitations. The key idea is to teach a LM to generate high probabilities for label tokens and low probabilities of negative candidates. Comprehensive experiments on language modeling and open-domain dialogue generation tasks show that the proposed contrastive token objective yields much less repetitive texts, with a higher generation quality than baseline approaches, achieving the new state-of-the-art performance on text degeneration.

## 1 Introduction

Autoregressive language models (LMs), such as OpenAI GPT-3 [1], have achieved impressive results on various natural language processing (NLP) tasks. The goal of training LMs is to learn the true distribution of a text corpus, and this is usually achieved through next word prediction. Specifically, a standard approach to training LMs is to minimize the cross-entropy loss between the true distribution and the model prediction. Unfortunately, LMs trained using the cross-entropy objective have been observed to exhibit text degeneration problems, where token, phrase, and sentence level repetition is a common symptom [6, 9, 27]. Such repeated texts differ markedly from those generated by humans.[1] To analyze the reasons for degeneration, our work views the vocabulary of LMs as being composed of three sets of tokens at each time step, i.e., positive tokens (label tokens), negative tokens (incorrectly repeating tokens), and irrelevant tokens (all the others). Based on this taxonomy, we stress that cross-entropy is in fact a contrastive learning objective that contrasts positive tokens with negative and irrelevant tokens. While it is necessary for LMs to learn how to rank positive tokens higher than other tokens in the predicted distribution, negative tokens are treated equally to irrelevant tokens (whose number is usually much larger) by the cross-entropy objective. As a consequence, negative tokens may not be suppressed hard enough.

To address the above issue, Welleck et al. [27] have proposed *unlikelihood training* to penalize certain negative tokens, i.e., tokens being incorrectly repeated. The key idea behind unlikelihood training is to lower the probability of negative tokens assigned by LMs. Despite its success, the unlikelihood objective penalizes negative tokens by decreasing their predicted probability but does

---

[1]Readers are referrred to Table 4 for some concrete examples. The degeneration problem even exists in large-scale, state-of-the-art, pre-trained language models such as GPT-3 [18].

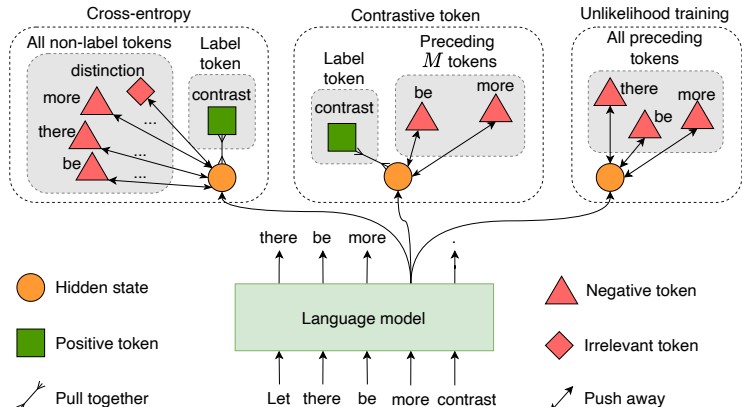

Figure 1: Illustrating the differences between our proposed contrastive token learning, unlikelihood training, and the cross-entropy objective for LMs. For contrastive token learning, we use the label token as the positive token and the preceding $M$ tokens as the negative tokens at each decoding step.

not consider the relationship between positive and negative tokens. Unlikelihood training also unintentionally boosts the probability of other irrelevant tokens. Moreover, all previous context tokens are used as negative candidates per generation step. Such an objective not only introduces a considerable amount of noise, but also results in sub-optimal repetition reduction, thus affecting the final generation performance.

In this paper, we introduce a simple yet effective *contrastive token learning* (CT for short) objective that integrates the best of cross-entropy and unlikelihood training, penalizing negative tokens by contrasting them with positive tokens. The commonalities and differences between cross-entropy, unlikelihood training, and CT are illustrated in Figure 1. Briefly, (i) without distinguishing between negative and irrelevant tokens, cross-entropy cannot effectively suppress negative tokens; (ii) due to the lack of contrast between negative and positive tokens, it is difficult for unlikelihood training to penalize negative tokens; and (iii) through its more focused contrast between positive and negative tokens, CT can take goal-directed actions rather than just predicting label tokens, i.e., explicitly teaching the LM to assign negative tokens with a lower probability than positive tokens. In this work, we combine the CT and cross-entropy objectives to train LMs, where cross-entropy performs on the label tokens so that they are assigned the highest probability, and CT effectively suppresses negative tokens from being generated.

We perform evaluations on the tasks of language modeling and open-domain dialogue generation.[2] Our empirical evidence demonstrates that LMs trained with the proposed CT objective can generate much less repetitive texts using standard greedy or beam search and achieve superior text generation performance under both automatic and human evaluations. CT has a minor negative influence on the perplexity of LMs, but thanks to the reduced repetition rates, in our case studies we observe substantial improvements regarding the quality of generated text.

## 2   Background

LMs aim to learn the true distribution over variable-length text sequences in a text corpus $X = (x_1, x_2, \ldots, x_{|X|})$ with $|X|$ tokens. A popular approach to this task is next word prediction, i.e., predicting a distribution over the next word following a given context. To train such a language model, cross-entropy and unlikelihood training are two representative objectives. In this section, we first review cross-entropy and unlikelihood training. We then provide an analysis of the text degeneration problem.

---

[2]Our source code, including data pre-processing scripts, our trained models, and an interactive Google Colab notebook, is available at `https://anonymous.4open.science/r/lit-seq`.

Table 1: The influence comparison of different learning objectives over the positive (label), negative (incorrectly repeating), and irrelevant tokens (all the others) for the LMs.

| Loss | Relevant tokens | | Irrelevant tokens | Contrast |
|---|---|---|---|---|
| | Positive | Negative | | |
| Cross-entropy (CE) | Promote | Suppress | Suppress | Yes |
| Unlikelihood training (UL) | Promote | Suppress/Promote | Promote | No |
| Contrastive token (CT) | Promote | Suppress | Unchanged | Yes |

## 2.1 Cross entropy

A standard approach to training a LM is to minimize the expected cross-entropy loss between the true distribution and the model prediction [28]. Specifically, the cross-entropy loss for each time step $t$ is defined as:

$$\mathcal{L}_{CE}^t = -\log p(x_t|x_{<t}) \tag{1}$$

$$= -\log \frac{\exp(h_t^T W_{x_t})}{\sum_{\hat{x}_t \in V} \exp(h_t^T W_{\hat{x}_t})} \tag{2}$$

$$= \log \left( 1 + \sum_{\hat{x}_t \in V, \hat{x}_t \neq x_t} \exp(h_t^T W_{\hat{x}_t} - h_t^T W_{x_t}) \right), \tag{3}$$

where $h_t$ is the model hidden state at time $t$, $W$ is the embedding matrix, and $W_{x_t}$ denotes the word embedding of token $x_t$. Through some simple transformations from Eq. (1)–(3), we can see that Eq. (3) is similar to the $N$-pair contrastive loss [24] for visual object recognition. In other words, cross-entropy effectively trains LMs to contrast the label tokens (positive examples) $x_t$ with all the other non-label tokens (negative and irrelevant examples) $\hat{x}_t \in V, \hat{x}_t \neq x_t$ in the whole vocabulary.

## 2.2 Unlikelihood training

To address the repetition issue of cross-entropy, Welleck et al. [27] have proposed unlikelihood training to penalize the likelihood of negative tokens (UL-T). The unlikelihood loss for time step $t$ is defined as:

$$\mathcal{L}_{UL}^t = -\sum_{x_t^- \in C^t} \log(1 - p(x_t^-|x_{<t})), \tag{4}$$

where $C^t = \{x_1, \ldots, x_{t-1}\} \backslash \{x_t\}$ is the set of negative tokens at time $t$, i.e., all previous context tokens. In this paper, we refer to this set of negative tokens as the *preceding tokens set*. As we will see in §2.3, UL-T does not work well as it can increase the probability of irrelevant tokens. Welleck et al. [27] have also proposed a more effective *sequence-level unlikelihood objective* (UL-S) that uses unlikelihood on decoded continuations during training time. We omit the details here as our proposed CT is more closely related to UL-T, but we do compare CT to UL-S in our experiments.

## 2.3 Discussion

The main difference between Eq. (3) and the $N$-pair contrastive loss is that, in Eq. (3), negative and irrelevant tokens are treated equally by cross-entropy.[3] These negative tokens need to be penalized harder than irrelevant tokens, otherwise, negative tokens may be incorrectly repeated in later time steps. This explains why LMs trained by cross-entropy have high repetition rates.

Although UL-T penalizes negative tokens, it does not work well enough, and as can be seen from Table 1, the reasons are twofold. First, each negative token is not definitely penalized because it depends on the influence of other negative tokens, which can be seen from the gradient analysis of UL-T (Eq. (11) in Appendix D). Second, the formulation of UL-T unintentionally boosts the probability of other irrelevant tokens and may make them surface as repeated tokens. We detail this analysis in §3.3.

---

[3]Albeit with different strengths, as seen in Eq. (10) in Appendix D.

## 3 Method

To address the issues discussed above and inherit the advantages of cross-entropy and unlikelihood training, in this section, we present a novel contrastive token learning (CT) objective for LMs. We first define the CT loss for each time step. Then we introduce a positive and negative token selection strategy. Finally, we discuss the differences and connections of CT with respect to cross-entropy and unlikelihood training.

### 3.1 Contrastive token learning

The key idea of CT is to promote positive (label) tokens in the ranking at each step, while lowering negative (incorrectly repeating) tokens, and leave other irrelevant tokens untouched. To this end, we formulate the CT loss for step $t$ as:

$$\mathcal{L}_{CT}^t = \log \left( 1 + \sum_{x_t^- \in S_N^t} \exp(h_t^T W_{x_t^-} - h_t^T W_{x_t}) \right),$$

(5)

where $S_N^t$ is the negative token set and $x_t$ is the positive token (i.e., label token) at time $t$. We detail the token selection mechanism of $S_N^t$ below.

During the training phase, we combine the CT loss with the cross-entropy loss for each time step as follows:

$$\mathcal{L}^t = \mathcal{L}_{CE}^t + \mathcal{L}_{CT}^t,$$

(6)

where $\mathcal{L}_{CE}^t$ aims to promote label tokens, training models to assign the highest probabilities to such tokens. On the other hand, $\mathcal{L}_{CT}^t$ focuses on contrasting positive tokens and negative tokens, so that the LMs can learn to effectively rank negative tokens lower than their positive counterparts.

### 3.2 Negative token selection strategy

Following [27], we use the *preceding tokens set* without requiring additional supervision as our negative tokens $S_N^t$. However, using all preceding tokens (as in [27]) may bring too much noise to the training process, especially for later time steps in a sequence. Hence, we instead propose to use the *preceding $M$ tokens set* to decide the negative tokens, with $M$ being a hyper-parameter. The set $S_N^t$ is defined as:

$$S_N^t = \{x_{t-M}, \ldots, x_{t-1}\} \backslash \{x_t\}.$$

(7)

Another difference with the *preceding tokens set* [27] is that, $S_N^t$ is a *multiset* that does not remove redundant occurrences. Intuitively, minimizing the CT loss with the *preceding $M$ tokens set* makes more frequently repeated tokens less likely to be predicted.

### 3.3 Gradient analysis

To see how loss functions influence the positive, negative and irrelevant tokens during training, we derive the gradient functions of each loss function with respect to these tokens in Appendix D. Table 1 is an intuitive summary of the influences, from which one can observe that: (i) Cross-entropy trains to promote label tokens in rankings at each time-step, while suppressing all the other tokens including negative and irrelevant tokens. (ii) It cannot be decided for unlikelihood training whether the negative tokens are promoted or suppressed by the gradient function (cf. Eq. (11) in Appendix D, the valid region for the corresponding gradient function contains both positive and negative values), and irrelevant tokens are promoted, both of which are problematic. (iii) With contrastive token learning, CT promotes positive tokens and suppresses negative tokens, and it is the only objective that does not affect irrelevant tokens (cf. the gradient functions in Appendix D).

When using CT together with CE, as we do for our final loss function, negatives are suppressed both in CT and in CE, while irrelevant tokens are only suppressed in CE. Therefore, our CT objective is able to better restrain incorrectly repeated tokens.

## 4 Related work

We review two lines of related work, i.e., neural text degeneration and contrastive learning.

**Neural text degeneration.** With large-scale pre-training, state-of-the-art neural LMs are able to generate human-like texts [1, 28]. However, they suffer from the *text degeneration problem*, where model-generated texts are dull and repetitive [6, 7, 27]. The text degeneration problem is especially serious with open-ended generation tasks, such as dialogue generation [9, 23] and language modeling [6, 27]. Some decoding approaches have been proposed to address this problem, by introducing randomness [4, 6] or disparity [23, 25] at inference time. Some other work suggests that the degeneration problem is caused by defects of the likelihood training objective, and improved training objectives have been proposed [8, 25, 27].

Our proposed contrastive token learning approach belongs to the training objective family. Compared to unlikelihood training [27], we address the suppression of repetitive tokens by contrasting them with positive tokens.

**Contrastive learning.** In computer vision, contrastive learning has been widely employed to learn representations [2, 10, 24]. Noise-contrastive estimation [5] has been proved successful for training word embeddings [16]. In recent years, contrastive learning has gained more attention in the area of natural language processing too. Most work builds contrasts at the sequence or document level by corrupting the ground truth sequence [3, 12, 14, 29] or mining positive/negative samples [17, 19].

Existing token-level contrastive learning frameworks contrast model representations from different positions [25, 30]. Differently, we contrast word embeddings while using the hidden representations as anchor points similar to the triplet contrastive loss [22]. Our formulation effectively contrasts logits output by the model for positive and negative tokens, thus it is more direct than unlikelihood training on addressing the repetitive degeneration problem. To the best of our knowledge, our proposed contrastive token learning is the first to use token embeddings as positive/negative examples in a contrastive framework for the text degeneration problem.

## 5 Experimental setup

We compare CT with baseline approaches on the language modeling and open-domain dialogue generation task. Since our experimental results on the dialogue task show a similar pattern as on the language modeling task, we will focus on the language modeling task in the body of the paper and postpone the setup and analyses of the dialogue task to Appendix I.

**Baselines and implementation.** We implement several state-of-the-art baselines and use them with GPT-2 [20]: (i) The vanilla cross-entropy (CE) objective; (ii) decoding-based methods: banning 3-grams [21], top-$k$ sampling [4], nucleus sampling [6] and contrastive search (SimCTG-CS) [25]; and (iii) learning-based methods: unlikelihood training [27], SimCTG [25], and noise-contrastive estimation (NCE; detailed in Appendix C) [5]. More details can be found in Appendix E.

**Dataset, training and inference details.** At training time, we fine-tune GPT-2 small on the widely-used Wikitext-103 dataset [15] with each learning-based approach (including the CE baseline) for 50K steps with 3K warm-up steps. As suggested in [27], for sequence-level unlikelihood training, we first fine-tune the language model using UL-T for 48.5K steps, and then switch to the UL-S objective for another 1.5K steps, resulting in UL-TS. Best model checkpoints for each task are selected according to the lowest validation CE loss with an evaluation interval of 1K training steps. We use trunks of 512 tokens, and a training batch size of 4. All models are trained using the Adam optimizer [11] with a learning rate of 1e-5. For UL-TS, we had to use a smaller learning rate of 1e-6, otherwise the generated texts contain massive ungrammatical repetitions (continuous token repetitions, as can be seen in Table 5 of Appendix F).

At inference time, we compare the performance of each approach to text degeneration using both greedy search and beam search. We use $k = 50$ for top-$k$ sampling, and $p = 0.9$ for deciding the sampling pool of the nucleus method. We follow Welleck et al. [27] to use 50 tokens as the input prefix and let the model generate 100 tokens as a continuation.

**Evaluation metrics.** We measure the perplexity (`ppl`) of different approaches. For measuring generative repetition, we follow Welleck et al. [27] to use 1-gram to 4-gram repetition rates (`rep-1` – `rep-4`), which are defined as the number of repeated $n$-grams divided by the total number of generated $n$-grams in each sequence, micro-averaged over the whole dataset. We also report the generation diversity at the dataset level, which is measured by distinct 1-gram rates (`dist-1`) [13] and unique 1-gram counts (`uniq-1`). We adopt human evaluation for measuring the quality of

Table 2: Results on the test set of Wikitext-103 for the language modeling task. ↑/↓ arrows denote whether higher or lower is better for a metric. The best result for either type of approach (decoding-based vs. learning-based) under each metric is highlighted in **bold face**. ‡ Does not count as the best. † For this experiment, we use a beam size of 5 as suggested in its original paper [25].

| | | ppl↓ | ppl-s↓ | search | rep-1↓ | rep-2↓ | rep-3↓ | rep-4↓ | dist-1↑ | uniq-1↑ |
|---|---|---|---|---|---|---|---|---|---|---|
| | GPT-2 | 18.01 | 25.95 | greedy | 71.03 | 60.12 | 54.77 | 50.93 | 1.15 | 12787 |
| | | | | beam | 77.02 | 69.70 | 65.49 | 61.69 | 1.12 | 12545 |
| *decoding-based* | 3-gram ban | 18.01 | 25.95 | greedy | 50.09 | 18.31 | *0.00‡* | *0.00‡* | 1.52 | 16940 |
| | | | | beam | 40.91 | 10.40 | *0.00‡* | *0.00‡* | 1.35 | 15114 |
| | Top-$k$ | 18.01 | 25.95 | greedy | **34.80** | **9.38** | **3.86** | **1.73** | **2.23** | **24840** |
| | | | | beam | 73.47 | 64.38 | 59.31 | 54.88 | 1.19 | 13280 |
| | Nucleus | 18.01 | 25.95 | greedy | 38.41 | 12.10 | 5.50 | 2.78 | 2.06 | 23038 |
| | | | | beam | 74.28 | 65.70 | 60.86 | 56.58 | 1.17 | 13004 |
| | SimCTG-CS | 18.12 | 26.10 | greedy | 70.23 | 58.92 | 53.44 | 49.54 | 1.17 | 13005 |
| | | | | beam† | **31.93** | **6.52** | **2.23** | **0.94** | **1.77** | **19746** |
| *learning-based* | SimCTG | **18.12** | **26.10** | greedy | 70.23 | 58.92 | 53.44 | 49.54 | 1.17 | 13005 |
| | | | | beam | 75.87 | 68.02 | 63.54 | 59.52 | 1.15 | 12835 |
| | NCE | 18.60 | 32.88 | greedy | 57.23 | 41.59 | 35.50 | 31.75 | 1.32 | 14774 |
| | | | | beam | 56.02 | 40.99 | 34.73 | 30.48 | 1.28 | 14322 |
| | UL-T | 18.93 | 26.63 | greedy | 60.91 | 45.15 | 38.31 | 33.90 | 1.26 | 14071 |
| | | | | beam | 67.39 | 55.95 | 49.85 | 44.78 | 1.15 | 12874 |
| | UL-TS | 18.88 | 27.41 | greedy | 51.98 | 29.17 | 19.71 | 14.42 | 1.29 | 14378 |
| | | | | beam | 45.81 | 23.96 | 15.60 | 10.41 | 1.27 | 14141 |
| | CT | 18.72 | 64.01 | greedy | **22.09** | **4.02** | **1.49** | **0.80** | **2.05** | **22832** |
| | | | | beam | **27.18** | **9.71** | **5.73** | **3.77** | **1.68** | **18697** |
| | Human | – | – | – | 29.92 | 7.25 | 2.81 | 1.14 | 3.41 | 19034 |

model generated texts. We randomly select 100 prefixes from the test set of Wikitext-103, and compare the continuations generated using CT with those by the best-performing baselines according to the automatic evaluation results. Since it does not make much sense to compare continuations with either side having excessive repetitions, we filter out such pairs using a threshold of rep-4 $\leq 0.05$ to make the comparisons more competitive. Then we display the prefix and two continuations from different systems (side-by-side, in a random order) to three crowd workers and ask them to select the winner in terms of repetition, coherence, fluency, and overall quality. Ties are allowed for all aspects. We use majority voting to decide the final winner. Details about our question form design and the instructions to crowd workers can be found in Appendix G.

## 6 Evaluation results

We conduct extensive experiments to demonstrate the advantages of our proposed CT. In this section, we discuss how CT compares to SOTA methods under both the automatic and human evaluations as well as showing some visualization analysis on its generation probability.

### 6.1 Baseline comparison

The performance comparisons between our CT and the baselines on the language modeling task are shown in Table 2. For models, the repetition and diversity results are calculated on model-generated continuations of 100 tokens, using 50 tokens of human-created text as the prefix. For the human performance, we calculate the metrics on trunks of 100 tokens for a fair comparison. The ppl metric is for 512-token sequences to comply with the training sequence length. To be comparable to existing work [25, 27], we also report ppl-s for short sequences of 50 tokens. We use a sequence length of 150 tokens and $M = 60$ as the negative window size for CT. Justifications for such hyper-parameter selections can be found in Appendix F.2.

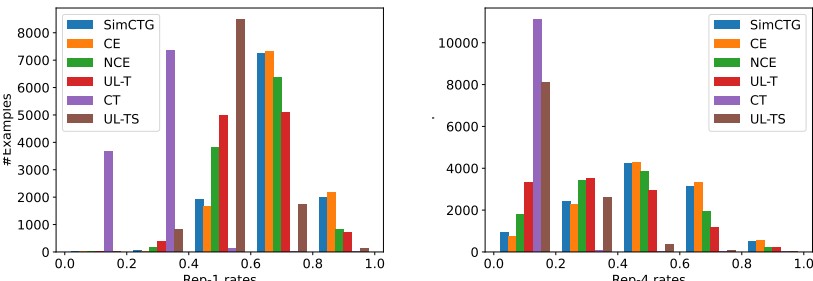

Figure 2: Histograms for `rep-1` (left) and `rep-4` (right) rates of each method, on the Wikitext-103 test set.

**CT compared to learning-based approaches.** One can observe that CT performs the best and even outperforms humans according to `rep-*` rates and unique token counts (`uniq-1`) when using greedy search. However, the repetition problem is still *not* yet solved, because when looking at specific cases, models trained by CT still occasionally generate texts with excessive repetitions, though being much rarer than baseline methods. To see how each method performs at every repetition level, we group the `rep-1` and `rep-4` rates of model-generated texts in to 5 bins, and plot their histograms in Figure 2, from which we can see that CT generates substantially less degenerated continuations (with `rep-1`$\geq$ 0.4 and `rep-4`$\geq$ 0.2). For UL-TS, we were able to achieve lower repetition rates with a larger learning rate of 1e-5 during training. However, the trained LM often generates ungrammatical repetitions. This problem does not exist with CT when trained with a learning rate as large as 1e-4. The comparisons are shown in Table 5 in Appendix F, and in §6.3 we show that this is caused by UL-TS being uncertain about its predictions at later time steps.

The diversity improvements brought by CT are the largest among all learning-based methods, especially when using greedy search. CT increases the second highest `uniq-1` count (NCE) by 55%. When comparing NCE and UL-T, one can see that utilizing the contrast between positive and negative tokens works better than solely penalizing negative tokens. The primary difference between CT and NCE is that the positive and negative tokens of CT *interact* with each other, while those of NCE do not (Table 1, more details in Appendix D). This explains the lower `rep-*` rates and higher diversity of CT, which also concurs with the observation made by Sohn [24] that interactive contrastive losses work better than non-interactive counterparts.

The `ppl` increase brought by CT is minor, with 0.71 points. When calculated on short sequences, due to the length mismatch of training and test sequences, `ppl-s` scores are higher than `ppl` for all approaches. Among them, contrastive objectives (NCE and CT) have larger `ppl-s` increases than other methods. Although CT has the highest increase on `ppl-s`, our case study (Table 4) shows that the generation quality of CT is not harmed, but on the contrary is improved due to the lower repetition and higher diversity of the generated texts.

**CT compared to decoding-based approaches.** Although CT is a learning-based method, we still compare it against decoding approaches for a more comprehensive understanding of its performance. When greedy search is used, CT outperforms the best decoding method (Top-$k$) in terms of `rep-*` rates, which again proves the effectiveness of contrastive learning. When using beam search, all but SimCTG-CS perform significantly worse than CT, both in terms of repetition rates and diversity. SimCTG-CS is effective at reducing repetition as it explicitly requires a disparity among different time steps at inference time. This can harm the generation quality, especially the coherence and fluency, as we see in §6.2. It is also worth noting that SimCTG-CS only works together with its SimCTG training objective and with beam search [25]. In summary, one can see that the repetition problem can be better addressed from the model learning perspective, in which case a simple greedy decoding strategy suffices.

## 6.2 Human evaluation

Human evaluation results are shown in Table 3. Regarding the overall quality, CT performs significantly better than Top-$k$ and SimCTG-CS, two decoding based approaches. Instead of purely learning generation policies from data, decoding approaches exert heuristics at inference time, which

Table 3: Win/lose rates (%) of CT compared to baselines under human evalutaions. For a competitive comparison, we filtered out highly repetitive examples of either model in the pair. * indicates statistical significance as determined with a sign test ($p < 0.05$).

| Comparison | Overall | | Repetition | | Coherence | | Fluency | |
|---|---|---|---|---|---|---|---|---|
| | Win | Lose | Win | Lose | Win | Lose | Win | Lose |
| CT vs Top-$k$ | 58* | 36 | 40* | 23 | 56* | 36 | 45 | 36 |
| CT vs SimCTG-CS | 55* | 35 | 46* | 18 | 52 | 36 | 54* | 28 |
| CT vs UL-TS | 48 | 43 | 43 | 28 | 39 | 45 | 47 | 38 |
| CT vs Human | 27 | 67* | 30 | 35 | 23 | 67* | 27 | 57* |

may prevent the language model from performing naturally. This explains the worse performance of decoding approaches on coherence and fluency. CT performs generally better than UL-TS except on coherence, but none of these differences are statistically significant. This suggests that CT has a similar generation quality as UL-TS on low-repetitive examples, but CT has much lower repetition rates as reported in Table 2. This result is expected, as both CT and UL-TS are learning-based approaches for training data-driven models, and on normal cases such as low-repetitive generations, they should perform similarly. Compared to human performance, there is still a large margin for machine learning models before they have a comparable performance on the language modeling task. Although CT performs on par with humans regarding repetition, its generations are far less coherent and fluent than those of humans. This may be mitigated by using larger models such as GPT-2 large or GPT-3. However, we could not perform such experiments due to a lack of computational resources.

### 6.3 Visualization analysis of the generation probability

We also conduct analyses to understand the predicted probability of model-generated tokens at inference time. As shown in Figure 3, diagonal cells represent the probability of generated tokens at the corresponding time steps; off-diagonal cells represent the probability of context tokens. The plots are averaged over 10 random instances from the test set of Wikitext-103.

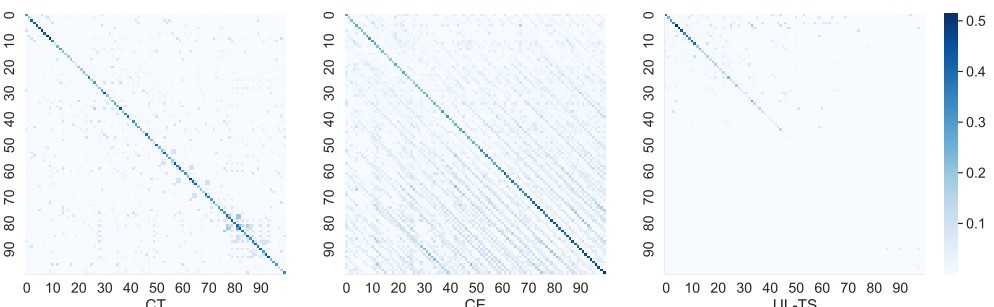

Figure 3: Heat maps for the generation probability of CT, CE and, UL-TS, at inference time. Row and column labels represent model-generated tokens at each time step, and the saturation of each cell represents the corresponding probability of each token. Please refer to §6.3 for a more detailed description. Heat maps for NCE, UL-T and SimCTG look similar to that of CE, and can be found in Appendix F, Figure 4.

We have the following key observations from Figure 3: (i) The heat map of CT shows a high variance in the diagonal, meaning that the model becomes certain and uncertain from time to time. As noted by Holtzman et al. [6], human-created texts also show such a pattern when fed through pretrained language models. (ii) In comparison, the heat map for CE shows clear stripes, which stand for excessive repetition of context n-grams. Besides, the diagonal cells are increasingly darker from top to bottom, revealing that the language model is becoming more and more certain about its later predictions, and it seems to positively correlate with the heavier repetition in the later halves of sequences. (iii) Contrary to CE, the heat map for UL-TS is almost white at the lower and the right parts of the heat map, indicating the language model is uncertain about any prediction in later stages, and the generated tokens just win marginally over other candidates. This is expected, since

Table 4: Continuations generated using UL-TS have heavier repetition than those generated using CT. Greedy search is used. More comparisons to other approaches can be seen in Table 6 in Appendix F.

| | | rep-1 |
|---|---|---|
| Prefix | the American lobster, H. americanus. It may grow to a length of 60 cm ( 24 in ) and a mass of 6 kilograms ( 13 lb ), and bears a conspicuous pair of claws. In life, the lobsters are blue, | |
| UL-TS | with a white spot on the underside. The eggs are laid in a single, white sac, and hatch in the spring. The adult is about **1 @.5 2** cm ( 0 @.@ 8 1 @.@ 9 in ) long, and weighs about **1 @.5 2** @.@ 5 kg ( 2 @.5 3 @.@ 8 lb ). The eggs are laid in a single, white @ brownish @ brown shell, with a white margin | 0.55 |
| CT | yellow, or greenish @-@ brown with short pointed teeth. The male lays eggs on top of the female's abdomen, which are incubated by means of tubes attached to the skin. After three weeks, the eggs hatch into adult males. = = Taxonomy = = The genus H. americanus has been described by several authors since its discovery in 1887. Most notably John Browning ( 1827 - 1901 ), who wrote about it in his book On Lobsters | 0.18 |
| Prefix | He married around 752, and by 757 the couple had had five children  three sons and two daughters  but one of the sons died in infancy in 755. From 754 he | |
| UL-TS | was a prominent figure in the Church of England, and in 758 **he was appointed vicar of** St John's, and in 759 **he was appointed vicar of** the Church of Scotland. He died in 761, and was succeeded by his son, John, who died in 762. = = Early life = = John was born **in the parish of St John's, in the parish of St John's, in the parish of St John's** parish, | 0.61 |
| CT | continued to live at St. John's Church, near Bath, where he received instruction from William de Montfort on how to build a dam for irrigation purposes. The first mention of this work came in 757 when it was discovered that a large portion of the earth beneath the riverbed had been washed away by floods caused by wind gusts. This led to speculation that it might be connected to the Norman invasion of England. In 758, however, Henry VIII granted permission for construction of a | 0.21 |

UL-TS penalizes repetitions unilaterally, and repetitions are more common in the later half of a model-generated sequence. Even though UL-TS is able to effectively reduce repetition rates, its heat map shows that the language model trained by UL-TS may subject to frequent grammatical errors, as can be seen in Appendix F, Table 5.

## 6.4 Case study

To intuitively see how well CT performs, we selected some example generations of CT, and compare them with those generated using UL-TS in Table 4. More often than not, continuations generated by CT are less repetitive and make more sense than those generated by UL-TS. The reason for the poor quality of UL-TS is that sequence-level unlikelihood training penalizes repeated 4-grams *generated* by LMs, making LMs uncertain about their predictions as suggested in Figure 3.

## 7 Conclusion and discussion

In this paper we studied the neural text degeneration problem. By integrating the best of cross-entropy and unlikelihood training objectives, we obtain a simple and effective contrastive token learning (CT) framework. The main novelty of this work is adapting contrastive learning to the token level of autoregressive language model training. As far as we are aware, our work is the first to use model hidden states as the anchor points and tokens as the positive and negative examples to formulate the contrastive loss. By contrasting the preceding $M$ tokens at a training step with the label token, LMs learn to not repeat such tokens, thus alleviating the repetition problem. Although the idea of negative tokens is similar to UL, our formulation of contrastive objective is more effective and safer to use. Experiments on the open-ended text generation and open-domain dialogue generation tasks show that CT beats UL-TS, the previous state-of-the-art approach to tackling the repetitive text degeneration problem. CT not only achieves the lowest repetition rates and the highest generation diversity, but also higher generation quality according to our human evaluation.

We performed experiments on fine-tuning LMs for reducing their repetition rates, which can be beneficial for related tasks such as abstractive summarization, machine translation, and image captioning. Our early experiments show that CT can be safely integrated when training a language model from scratch, which can be helpful for future pre-training of large language models. In this work, we used CT with decoder-only (GPT2) and encoder-decoder (BlenderBot) language models, but we note that CT can also be used with encoder language models (e.g., BERT [26]) to potentially improve the model performance such as prediction accuracy. The repetitive degeneration problem is still not fully solved as occasional, excessive phrase repetitions remain in the generated texts. We leave these research directions as future work.

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
