# OpenReview forum: "A Simple Contrastive Learning Objective for Alleviating Neural Text Degeneration"
_NeurIPS.cc/2022/Conference — NeurIPS 2022 Submitted_

### Official Review · Reviewer_aoYo · 2022-07-11

**Rating:** 3
**Confidence:** 5
**Soundness:** 1 poor
**Presentation:** 3 good
**Contribution:** 3 good

**Summary:**

This paper propose a learning-based method for alleviating the problem of text degeneration.  The proposed method is simple. I combines unlikelihood training and contrastive learning.  The training objective is to promote label tokens while suppressing preceding tokens. Experimental results show that the proposed method can reduce repetitions in generated text.

**Questions:**

My major concern: for rep-1/2/3/4, dist-1, uniq-1, I think the best results should be the ones that are closest to human-generated text. if so, you have to redo the analysis of experiment results.

**Strengths And Weaknesses:**

Strengths:
1. The method is simple.
2. The experiments are elaborate.
3. The proposed method seems effective in reducing repetitions.

Weaknesses:
1. The idea to penalize any repetitions in a short window is somewhat problematic. Some repetitions are reasonable in human-generated text. I think the critical question is how to properly construct the negative token set.
2. The analysis of experimental results is problematic. For rep-1/2/3/4, dist-1, uniq-1, I think the best results should be the ones that are closest to human-generated text. Like the above, humans generate repetitions naturally. NOT all repetitions are evil.

---

> ### Author Response · Authors · 2022-08-02
> **Response to Reviewer aoYo**
>
> Thank you for your valuable comments. Please see below our responses.
>
> > The idea to penalize any repetitions in a short window is somewhat problematic. Some repetitions are reasonable in human-generated text. I think the critical question is how to properly construct the negative token set.
>
> There appears to be a misunderstanding here. We already considered the scenario of reasonable repetitions, and that is why we constructed our negative token set to exclude natural repetitions. Please refer to Equation (7). Or did you mean something else?
>
> > My major concern: for rep-1/2/3/4, dist-1, uniq-1, I think the best results should be the ones that are closest to human-generated text. if so, you have to redo the analysis of the experiment results.
>
> Thanks for pointing this out. This comment concurs with one from Reviewer QKnA. We agree that the repetition rates are not the lower the better. In the current version, we decided the best hyper-parameter settings by optimising the repetition rates. But through the parameter sweeping experiments in Appendix F.2, we can see that the repetition rates of our method are highly responsive to M, from which we can easily tune the repetition rate to be close to human performance. Please see below such a set of results. We will add them and their discussion to the revised paper.
>
> ||ppl|ppl-s|search|rep-1|rep-2|rep-3|rep-4|dist-1|uniq-1|
> |-|-|-|-|-|-|-|-|-|-|
> | CT-Preced-30| 18.67 | 52.77 | greedy | 26.74 | 8.23 | 3.73 | 1.52 | 1.93 | 21562 |
> |||| beam| 31.13 | 13.66 | 9.28 | 7.00 | 1.61 | 18016 |
> |Human| - | - | - |  29.92 | 7.25 | 2.81 | 1.14 | 3.41 | 19034 |

---

> > ### Comment · Reviewer_aoYo · 2022-08-10
> > **answer**
> >
> > - You only exclude the current ground truth. This is totally different to "exclude natural repetitions". There could be other reasonable repetitions other than the current ground truth.
> > - This is not only about hyper-parameters tuning. This is about the correct evaluation of generated texts. You do not use the metrics correctly. To me, this is a fundamental flaw in this paper and you should redo all analysis.

---

> > > ### Author Response · Authors · 2022-08-10
> > > **Thanks for your constructive communication**
> > >
> > > > You only exclude the current ground truth. This is totally different to "exclude natural repetitions". There could be other reasonable repetitions other than the current ground truth.
> > >
> > > We agree that this can be a valid concern, although we think providing some concrete examples might be helpful for us to understand your concern. However, we would like to point out two major misunderstandings in this argument:
> > > 1. Since our proposed CT is closely related to CE, your concern can be equally valid for CE. This work focuses on addressing CE's ineffectiveness of reducing repetition, which is proven to be helpful by narrowing down the scope of selecting negative tokens.
> > > 1. Even if this is a valid concern, and it is a concern that is quantifiable and serious enough, although we doubt that, we would like to kindly point out that we are not introducing new problems. We would love to solve multiple problems in one go, but that is restricted by many other factors, such as the priority, time and resource restrictions, etc. We can add this to the discussion that this can be a future topic of investigation.
> > >
> > > > This is not only about hyper-parameters tuning. This is about the correct evaluation of generated texts. You do not use the metrics correctly. To me, this is a fundamental flaw in this paper and you should redo all analysis.
> > >
> > > We respectfully disagree with this argument. To us it is more of a preference of hyper-parameters. As we mentioned in the conclusion, even with our optimization for low repetition rates, there are still cases with excessive repetitions, which is devastating to the response quality [5]. With a more gentle anti-repetiton setting, this problem is more serious. Nevertheless, we agree that the claim of "outperforming humans" is incorrect, and will remove such claims in the revised manuscript.
> > >
> > > ---
> > > [5] Holtzman, Ari, et al. "The curious case of neural text degeneration."

---

> > > > ### Comment · Reviewer_aoYo · 2022-08-10
> > > > **answer**
> > > >
> > > > - Taking the sentence "I like apple and banana, apple ..." as an example, here apple is a reasonable repetition. In the mean time, banana could also be a reasonable repetition. Because the main idea of this paper is differentiating positive, negative and irrelevant tokens, to me, not considering this issue makes the paper weak. For this reason, I would not advocate for accept.
> > > > - This paper is concerned with text generation and the results are evaluated with a set of evaluation metrics. But it seems that the authors do not understand how to use these automatic metrics correctly (That is, the authors cannot correctly tell which one is better given the results under these metrics). It is like the goal is wrong, so I really cannot trust the experiments. For this reason, I lean towards reject.
> > > > - I believe the authors can fix the above points, at least the second point, in their next version.

---

> > > > > ### Author Response · Authors · 2022-08-10
> > > > > **Thanks for your answer**
> > > > >
> > > > > > Taking the sentence "I like apple and banana, apple ..." as an example, here apple is a reasonable repetition. In the mean time, banana could also be a reasonable repetition. Because the main idea of this paper is differentiating positive, negative and irrelevant tokens, to me, not considering this issue makes the paper weak. For this reason, I would not advocate for accept.
> > > > >
> > > > > Thanks for the example. In this case, "banana" is also suppressed by cross-entropy at the step of predicting the second "apple", which shows that this is not a new problem we introduced. Our contribution is to distinguish negative tokens from irrelevant tokens, which is shown to be more effective than cross-entropy on reducing repetition. But admittedly, our solution is not **perfect**. Thanks for pointing it out. We will add it to the discussion that this needs further investigation, but it is out of the scope of this work.
> > > > >
> > > > > > This paper is concerned with text generation and the results are evaluated with a set of evaluation metrics. But it seems that the authors do not understand how to use these automatic metrics correctly (That is, the authors cannot correctly tell which one is better given the results under these metrics). It is like the goal is wrong, so I really cannot trust the experiments. For this reason, I lean towards reject.
> > > > >
> > > > > First of all, all existing work use such metrics and optimise for both lower ppl and repetition rates, such as [3, 5-7]. Second, this criticism can be applied to almost any other automatic evaluation metrics, such as BLEU, ROUGE, and perplexity, etc. A better automatic score does not always mean a better quality [8], and that is exactly what we mentioned in the paper that we rely on human evaluation for qualitative comparisons. It is unfortunate that we did not propose a more reliable evaluation metric, but again, this can not be concluded as the drawback of our work. Besides, for the case study in Table 4, we can see that the text generated using CT still contains a lot of stop words, such as "to" and "of", which we believe are "natural repetitions" you are talking about. Currently we do not have evidence that such natural repetitions are harmed by using CT.
> > > > >
> > > > > > I believe the authors can fix the above points, at least the second point, in their next version.
> > > > >
> > > > > As we mentioned in earlier comments, we will add such discussions that solely optimising for low repetitions might not always be wanted, and we will also remove the claims of "outperforming human" because from human evaluation results we clearly did not outperform human.
> > > > >
> > > > > ---
> > > > > [6] Su, Yixuan, et al. "A Contrastive Framework for Neural Text Generation."
> > > > > [7] See, Abigail, et al. "What makes a good conversation? how controllable attributes affect human judgments."
> > > > > [8] Liu, Chia-Wei, et al. "How not to evaluate your dialogue system: An empirical study of unsupervised evaluation metrics for dialogue response generation."

---

### Official Review · Reviewer_QKnA · 2022-07-11

**Rating:** 4
**Confidence:** 4
**Soundness:** 3 good
**Presentation:** 2 fair
**Contribution:** 3 good

**Summary:**

This paper introduces a simple contrastive objective for text generation models to alleviate text degeneration. The proposed objective contrasts the ground-truth tokens with the negative tokens (previous $M$ context tokens), thus suppressing the model from generating those tokens during inference. Empirical results show the proposed approach can reduce the repetition significantly.

**Questions:**

- In figure 3, I wonder what would UL-T look like? And do you think the uncertainty in the later steps can be solved by introducing $M$ to UL?

**Limitations:**

Yes

**Strengths And Weaknesses:**

**Strengths**
- The proposed approach is simple and intuitive.
- The proposed approach effectively reduces the repetition in the generated text.
- The analysis is extensive and interesting.

**Weaknesses**
- Some confusion about the description.
    + The authors keep claiming that Unlikelihood training (UL) does not consider the relationship between the label tokens and unlikely tokens but they never give details about how the proposed approach (CT) considers this.
    + I don't understand why UL does not contrast the negative tokens to other tokens. Essentially we are looking at the probability distribution. So when the probabilities for negative tokens are reduced because of UL, the probabilities for other tokens are relatively boosted compared to negative tokens. Is this not considered as contrastive?
    + The authors claim that the fact that UL considers all the previous context tokens as negative tokens introduces too much noise and results in sub-optimal repetition reduction. However, in the proposed approach, the same strategy is used to define negative tokens with the exception of an additional window of size $M$, which I believe can be applied to UL directly. And how is this not sub-optimal compared to UL?
    + The main intuition in this work is to dynamically promote/suppress tokens based on three categories (positive, negative, and irrelevant). Similar stuff has been observed in existing work [1] and it is very relevant to this work. The author should at least discuss this regard.
- Experiments
    + Hyper-parameters. How is the hyper-parameter used in UL? Apparently, if UL is applied more aggressively, the repetition could be reduced more obviously although the ppl could be harmed. However, there is no any description of the hyper-parameter choice of UL. Similarly, I cannot really agree with the claim that the CT outperforms the top-$k$/$p$ in terms of reducing repetition. Once a larger value of $k$ or $p$ is used, the repetition is almost guaranteed to be reduced. But here again, no different values of $k$ and $p$ are considered.
    + I disagree with the claim that CT even outperforms humans according to rep-* and unique token counts. The repetition is not always the less the better, and the number of unique tokens is not always the more the better, as the good text requires a certain level of repetition to retain the coherence.
    + Human evaluation. Human evaluation is somewhat biased and subjective. Even major vote is used, it will be more convincing to report some correlation between humans.
- Line 265: analyses --> analysis


[1] Lin et al., "Straight to the Gradient: Learning to Use Novel Tokens for Neural Text Generation"

---

> ### Author Response · Authors · 2022-08-02
> **Response to Reviewer QKnA**
>
> Thank you for your valuable comments. Regarding your confusion and questions:
>
> > The authors keep claiming that Unlikelihood training (UL) does not consider the relationship.. but they never give details about how the proposed approach (CT) considers this...
>
> This claim is supported by comparing Eq. (4) and (5). UL is defined solely by considering negative tokens, while CT loss is defined as the contrast between the logits of negative and positive tokens. We will make this more explicit in 3.1 after Eq. (5) of the revised manuscript.
>
> > I don't understand why UL... Is this not considered as contrastive?
>
> In our paper, we follow the previous works such as [2] for the definition of contrastive loss. In this definition, we refer to a direct comparison between negative and positive examples as “contrastive”. Since the UL loss is calculated solely by looking at the probabilities of negative tokens, it does not match the conventional definition of contrastive losses.
>
> > The authors claim that the fact that UL... And how is this not sub-optimal compared to UL?
>
> As UL, we have tried using all the preceding tokens for CT, which did not give us the best results. Due to the space limitation, we have put the experiments for checking the influence of M in Figure 6, Appendix F.2.
>
> As suggested, we tried smaller M values with UL-T, but did not see much difference. This is mainly due to that UL-T uses a token set while we use a multiset. When we use our preceding M token multiset, we achieve better repetition performance for UL-T. Please see the results above in our response to Reviewer TrP3.
>
> > ... Similar stuff has been observed in existing work [1]...
>
> Thanks for pointing out this work. It indeed shares a similar spirit with our work. We will add a discussion to the related work.
>
> > How is the hyper-parameter used in UL? ...
>
> Since we are using the same experimental setup as UL for the language modeling task, we followed the best hyper-parameter settings reported in the UL work [3], which is $\alpha=1$ for UL, k=50 for top-k and p=0.9 for top-p. We will make this clearer in section 5 for the revised version.
>
> As suggested, we tried larger $\alpha=\{2, 5, 10\}$ for UL and show their results below, indeed all achieve lower repetition rates but still underperform CT, and inevitably suffer from higher perplexity.
>
> ||ppl|ppl-s|search|rep-1|rep-2|rep-3|rep-4|dist-1|uniq-1|
> |-|-|-|-|-|-|-|-|-|-|
> | UL-T-$\alpha 2$ | 21.07 | 27.97 | greedy | 53.58 | 35.11 | 27.93 | 23.66 | 1.33 | 14885 |
> | UL-T-$\alpha 5$ | 29.43 | 32.72 | greedy | 42.09 | 21.17 | 14.43 | 11.01 | 1.47 | 16361 |
> | UL-T-$\alpha 10$ | 42.56 | 39.57 | greedy | 33.54 | 14.11 | 9.04 | 6.83 | 1.64 | 18310 |
>
> > I disagree with the claim that CT even outperforms humans...
>
> Thanks for pointing this out. Indeed the repetition rates are not the lower the better. In the current version, we decided the best hyper-parameter settings by optimising the repetition rates. But through the parameter sweeping experiments in Appendix F.2, we can see that the repetition rates of our method are highly responsive to M, from which we can easily tune the repetition rate to be close to human performance. Please see below such a set of results, and we will add their discussion to the revised paper.
>
> ||ppl|ppl-s|search|rep-1|rep-2|rep-3|rep-4|dist-1|uniq-1|
> |-|-|-|-|-|-|-|-|-|-|
> | CT-Preced-30| 18.67 | 52.77 | greedy | 26.74 | 8.23 | 3.73 | 1.52 | 1.93 | 21562 |
> |||| beam| 31.13 | 13.66 | 9.28 | 7.00 | 1.61 | 18016 |
> |Human| - | - | - |  29.92 | 7.25 | 2.81 | 1.14 | 3.41 | 19034 |
>
> > Human evaluation is somewhat biased and subjective...
>
> Below we calculated the agreement percentage following [1], and Krippendorff's $\alpha$ of the human annotations. The agreements are between 0.27-0.44, which is comparable to that of [4] (between 0.25-0.45).
>
> || 3 agree % | 2 agree % | disagree % | Krippendorff's $\alpha$ |
> |-|-|-|-|-|
> | CT vs Top-k| 37| 58| 5| 0.33|
> | CT vs SimCTG-CS | 26| 66| 8| 0.27|
> | CT vs UL-TS| 34| 60| 6| 0.28|
> | CT vs Human| 36| 58| 6| 0.44|
>
> > In figure 3, I wonder what would UL-T look like? ...
>
> As we mentioned in the caption, heatmaps for NCE, UL-T and SimCTG look similar to that of CE, and can be found in Appendix F, Figure 4. Introducing M to UL-TS might alleviate the uncertainty problem, but cannot solve the problem. This is because UL-TS uses already generated tokens as negative tokens, which can be any token. In the end, the model just learns to be uncertain of any token. We have added an experiment by using a small M=60 for UL-TS, and indeed the heatmap is similar to that of the original UL-TS.
>
> > Line 265: analyses --> analysis
>
> Thanks. Changed in the new version.
>
> ---
>
> [2] Chen, Ting, et al. "A simple framework for contrastive learning of visual representations."
>
> [3] Welleck, Sean, et al. "Neural text generation with unlikelihood training."
>
> [4] Adiwardana, Daniel, et al. "Towards a human-like open-domain chatbot."

---

> > ### Comment · Reviewer_QKnA · 2022-08-08
> > **Thanks for the responses**
> >
> > Thank you for the effort and time to provide additional experiment results.
> >
> > - In the responses to reviewer TrP3, I found it a bit counter-intuitive that using the proposed negative token set increases the **ppl** in UL. In my option, the reduction of the size of the negative token set (from all previous tokens to $M$) should decrease the impact the UL loss has on the original cross-entropy loss. Would you provide some insight here, e.g., what is the average size of the multi-set, etc?
> >
> > - I still don't think the comparison between the proposed approach and the decoding algorithm is fair, let alone the claim that the proposed approach outperforms them, i.e., only a single choice of $k$ ($k=50$) and $p$ ($p=0.9$) in top-$k$ and top-$p$ sampling.
> >
> > - I still don't think the claims/motivations (a. without distinguishing between negative and irrelevant tokens, cross-entropy cannot effectively suppress negative tokens; b. due to the lack of contrast between negative and positive tokens) from the paper are convincing.

---

> > > ### Author Response · Authors · 2022-08-08
> > > **Thanks for putting effort into understanding our work**
> > >
> > > Please see below our responses.
> > >
> > > > In the responses to reviewer TrP3, I found it a bit counter-intuitive that using the proposed negative token set increases the ppl in UL. In my option, the reduction of the size of the negative token set (from all previous tokens to $M$) should decrease the impact the UL loss has on the original cross-entropy loss. Would you provide some insight here, e.g., what is the average size of the multi-set, etc?
> > >
> > > There are mainly two reasons. First, due to the way UL calculates the loss, it does not definitely suppress the negative tokens, but **definitely** promotes irrelevant tokens, as we summarise in Table 1, and further theoretical proofs can be found in Eq. (11) and (12) of Appendix D. This can have a negative impact on the ppl, but from Table 1 and Eq. (13) - (15) of Appendix D, our CT does not suffer from this problem as irrelevant tokens are not influenced. Second, the use of multiset preserves repeated tokens, which magnifies the problem of UL because irrelevant tokens can be promoted for several times more. Again, since our CT does pair-wise contrast, it does not influence irrelevant tokens.
> > >
> > > > I still don't think the comparison between the proposed approach and the decoding algorithm is fair, let alone the claim that the proposed approach outperforms them, i.e., only a single choice of $k$ ($k=50$) and $p$ ($p=0.9$) in top-$k$ and top-$p$ sampling.
> > >
> > > We would like to reiterate that we used the same experimental setup of the LM task as the UL work, in which the authors tried several different parameters for top-$k$ and top-$p$ and we follow the best setting reported by that work [3]. Besides, we also would like to emphasize that these sampling methods have fundamental differences compared to learning-based methods such as CT. As pointed out in [3], sampling-based methods "do not address the fact that the token-level probabilities predicted by the model are poor". Moreover, the random sampling methods are subject to the gibberish problem [5], thus should be always used carefully, i.e., with small $k$ and $p$ for top-k and top-p methods. Considering the average token-level ppl of our fine-tuned models, which is around 19, meaning that at each prediction step, the LM is confused by about 3 tokens. In this regard, our selection of $k=50$ and $p=0.9$ are already quite aggressive, yet still did not produce satisfactory results, which is further supported by the worst performance of top-k according to our human evaluation in Table 3.
> > >
> > > > I still don't think the claims/motivations (a. without distinguishing between negative and irrelevant tokens, cross-entropy cannot effectively suppress negative tokens; b. due to the lack of contrast between negative and positive tokens) from the paper are convincing.
> > >
> > > From Table 1 and Eq. (9) and (10) of Appendix D, we can see that cross-entropy indistinguishably suppresses negative and irrelevant tokens, in which case the negative tokens are possibly **not** suppressed compared to irrelevant tokens. When we put direct contrast between negative and positive tokens, negative tokens are suppressed compared to both positive and irrelevant tokens.
> > >
> > > ---
> > > [5] Holtzman, Ari, et al. "The curious case of neural text degeneration."

---

### Official Review · Reviewer_TrP3 · 2022-07-13

**Rating:** 6
**Confidence:** 4
**Soundness:** 3 good
**Presentation:** 3 good
**Contribution:** 3 good

**Summary:**

This work proposes contrastive token learning in order to alleviate the degeneration problem in text generation tasks. In addition to the conventional token-wise cross entropy loss, the loss penalizes negative tokens in the similar manner as done in cross entropy loss, but sum  over negative tokens and the true token in the denominator. The negative tokens are chosen from the past history in a window but allowing duplicates by changing from a set to a multiset. Experiments are carried out on language modeling and dialogue generation tasks and the proposed method achieves lower repetition rates with more distinctive outputs preserving comparable perplexities when compared with SOTA baselines.

**Questions:**

None

**Limitations:**

* Currently, the proposed approach is experimented on language modeling and dialogue generation, but it is not clear whether the approach is also effective to conditional language model setting.

**Strengths And Weaknesses:**

Strengths
* The proposed method based on the unlikelihood training by penalizing negative tokens, but employs contrastive loss for softly penalizing negative tokens. The idea is simple yet sound.
* The use of tokens in a window and multiset sounds the major contribution in this work.
* Experiments are well designed and the effectiveness is demonstrated by the systematic comparison. Analysis is also convincing both qualitatively and quantitatively.

Weaknesses
* One of the main contributions is the carefully designed negative token set and the set could be easily employed in other approaches, e.g., UT and SimCTG. I'd like to see the comparisons with those SOTA approaches with the proposed negative token set.

---

> ### Author Response · Authors · 2022-08-02
> **Response to Reviewer TrP3**
>
> Thank you for your valuable comments. Please see below our responses.
>
> > One of the main contributions is the carefully designed negative token set and the set could be easily employed in other approaches, e.g., UT and SimCTG. I'd like to see the comparisons with those SOTA approaches with the proposed negative token set.
>
> Thanks for this suggestion. We have added some experiments using UL-T/SimCTG with our preceding M token set. Since SimCTG does not support multiset, below we analyse the results of UL-T. Preceding M tokens set with UL shows more obvious repetition reduction effects than vanilla UL, especially with a bigger M. This supports our employment of a multi-set that allows more frequently repeated tokens to be suppressed more heavily. However, even with our negative token set, UL still underperforms CT on repetition rates and suffers from higher perplexity. We have put below the best results regarding repetition rates for the UL-T method using preceding M=90 token sets, and we will add them with analysis to the revised manuscript.
>
> |               | ppl | ppl-s | search | rep-1 | rep-2 | rep-3 | rep-4 | dist-1 | uniq-1 |
> |---------------|-----|-------|--------|-------|-------|-------|-------|--------|--------|
> | UL-T-Preced-M-multiset | 21.18 | 27.74 | greedy | 50.01 | 31.52 | 24.46 | 20.30 | 1.41 | 15676 |
> |               |     |      | beam   | 52.99 | 38.00 | 30.78 | 25.19 | 1.28 |14318 |
>
> > Currently, the proposed approach is experimented on language modeling and dialogue generation, but it is not clear whether the approach is also effective to conditional language model setting.
>
> Thanks for the comment. Our dialogue generation task is actually using a conditional language model (encoder-decoder). Its results are in Appendix I and show similar effectiveness of CT as on the language modeling task, achieving the lowest repetition rates. We will make this message more explicit in the introduction and at the beginning of the experimental setup section.

---

### Meta-Review · Area_Chair_KdLz · 2022-08-29

**Recommendation:** Reject
**Confidence:** Certain

**Metareview:**

The paper proposes a simple contrastive objective punishing the degeneration in the text generation. There is an agreement in the reviewers that the method itself is somewhat incremental and lacks very clear justification. It could be seen as a straight-forward extension to the SimCTG algorithm and there have been a handful of publications in the field. I therefore do not recommend acceptance of this paper to NeurIPS.


**Award:**

No

---

### Decision · Program_Chairs · 2022-09-14

Reject